# Towards Fine-grained Safety Evaluation of Large Reasoning Language Models in Task Planning

## Abstract

Large Reasoning Language Models (LRLMs) show strong potential in robotic task planning, but their reasoning processes remain unreliable: they may violate safety constraints, reducing the likelihood of producing correct plans, or show inconsistencies between the reasoning process and final results, which undermines interpretability and user trust. Existing evaluations of LRLMs rely mainly on outcome-based metrics, such as task success rate and token efficiency, which fail to capture these critical reasoning properties. This gap is especially concerning in safety-critical planning domains, where verifying the correctness of reasoning is essential. To address this issue, we propose a fine-grained safety evaluation framework that systematically analyzes the reasoning processes of LRLMs in task planning problems. Our method segments reasoning into chunks, summarizes each chunk into explicit planning steps, and verifies them against safety constraints using an external verifier, while applying rollback techniques to prevent bias in subsequent reasoning. Using a dataset of Planning Domain Definition Language (PDDL)-based problems, we conduct extensive experiments on various LLMs and LRLMs. The experimental results reveal the inconsistency between the reasoning process and the final output of LRLMs, as well as their limitations in detecting and correcting safety violation errors in their own reasoning process. These findings point out directions for future improvements.

## 1 Introduction

Generating coherent sequences of executable actions for robotic task planning has long been envisioned as a key capability of Large Language Models (LLMs). Unlike traditional planners that rely on search-based algorithms (Sotirchos & Ajanovic, 2024; Karur et al., 2021; Baier et al., 2009; Zhu & Givan, 2005) or mathematical optimization techniques (Zhao et al., 2024; Dong et al., 2023; Janner et al., 2021; Kelly, 2017), LLM-based approaches offer greater flexibility for complex, dynamic, and open-ended tasks, as they leverage strong reasoning and natural language understanding to interpret high-level instructions and adapt plans to changing contexts or user feedback. Building on this foundation, the emergence of Large Reasoning Language Models (LRLMs), such as GPT-o1 (OpenAI, 2024) and DeepSeek-R1 (Guo et al., 2025), reflects a growing focus on enhancing reasoning capabilities, as these models are designed to produce structured multi-hop chains of thought with intermediate reasoning steps that ensure logical consistency, rather than merely predicting the next token based on surface-level patterns. Taking advantage of these reasoning capabilities, recent studies have shown that LRLMs achieve substantial performance gains over conventional LLMs on safety-critical domains, such as task planning.

Despite their promising reasoning abilities, LRLMs often exhibit reasoning processes whose safety cannot be guaranteed, which fundamentally limits their reliability in robotic task planning. In particular, the reasoning process may break important safety constraints in the planning task. Such violations can make intermediate steps invalid and greatly lower the chance of producing correct and safe final plans, which leads to a lower overall task success rate. Another common issue is the inconsistency between the reasoning process and the final results: LRLMs may sometimes produce correct plans while using unsafe reasoning process, which means the plans are not supported by reliable reasoning. This inconsistency also reduces interpretability and transparency. When some

reasoning chunks are wrong but the final plan looks correct, users cannot use the reasoning to check safety or find errors, which makes the system harder to trust. This lack of alignment between reasoning and results lowers user confidence in using LRLM-based planning systems in safety-critical settings. Moreover, it is also important to consider whether LRLMs can detect and correct their own earlier safety violations during the reasoning process. This self-detection and self-correction ability could, in principle, help enhance the safety of LRLM-driven systems, but its presence and effectiveness in current LRLMs remain unclear. Evaluating this capability is therefore crucial for understanding the reliability of their reasoning behavior.

However, current evaluations of LRLMs in the context of the task planning domain largely rely on metrics that were originally designed for LLMs, such as task success rate and token efficiency. These metrics (outcome-level) fail to capture the unique reasoning capabilities introduced by LRLMs and are also unable to assess the safety of the reasoning process. Some efforts in other domains that attempt to assess reasoning process mainly focus on semantic consistency, logicality, informativeness, and fluency, etc. (Golovneva et al., 2022; Opitz & Frank, 2020; Creswell et al., 2022; Leiter et al., 2022), yet these criteria (token-level) fall short in safety-critical contexts. To address this problem, it is necessary to develop comprehensive evaluation frameworks for assessing the safety of LRLMs in the task planning domain, with a particular focus on their reasoning process. The evaluation should cover not only correctness and consistency, but also their ability to perform self-correction during reasoning. Such fine-grained chunk-level safety evaluations would provide a foundation for deploying LRLMs as the brain of safety-critical robotic systems, guiding future efforts toward improving both their safety and task success rates.

A key challenge lies in the fact that these reasoning processes are expressed in natural language, which is often verbose, unstructured, and inherently ambiguous. Unlike formal logic representations, natural language reasoning steps cannot be directly verified against predefined safety specifications or logical constraints using existing automated verifiers, making systematic evaluation of their correctness and safety particularly difficult. A second challenge arises when moving from local evaluation of reasoning chunks to process-level evaluation. The reasoning process can be divided into multiple chunks, each containing a sequence of action steps. Even if the safety of individual chunks can be assessed, it is still unclear how to aggregate these results to evaluate the overall reasoning process. This requires new ways of analyzing how early safety violations influence later reasoning, whether the model is able to repair its own mistakes, and whether different chunks remain consistent with each other. Addressing these challenges is essential for building comprehensive evaluation frameworks that can guide the safe and reliable deployment of LRLMs in robotic task planning.

To address these challenges, we first propose a fine-grained verification framework to evaluate the safety of the LRLM reasoning process. Specifically, we segment the reasoning process of the LRLMs by detecting transition words and treating the text generated up to each transition as a distinct reasoning chunk. After each reasoning chunk is produced, we adopt an approach inspired by enforced prefixes to prompt the model to summarize the current reasoning chunk into a set of explicit planning steps. These intermediate plans are then passed on to an external verifier to verify their compliance with predefined safety constraints. Once each summary is generated and verified, techniques such as key–value (KV) cache rollback and input truncation are applied to remove any residual bias introduced by the summarization step, ensuring that the model can continue its reasoning process without being influenced by the intermediate summarization and verification. Beyond chunk-level verification, we further design a set of metrics to assess the safety of LRLM reasoning, including safety consistency, safety violation detection and correction, and token usage. To support these evaluations, we construct a dataset of task planning problems expressed in the Planning Domain Definition Language (PDDL).

Using this framework, we conduct extensive experiments on a range of LLMs and LRLMs, focusing on their reasoning processes, compliance with safety constraints, and capacity for self-correction. The experimental results reveal the inconsistency between the reasoning process and the final output of LRLMs, as well as their limitations in detecting and correcting safety violation errors in their own reasoning process. These findings point to promising directions for improving the safety and performance of LRLMs in safety-critical domains, such as task planning. We will release our dataset and the source code for generation and evaluation to facilitate future research.

## 2 RELATED WORK

### 2.1 LANGUAGE MODELS FOR TASK PLANNING

With the rise of language models, such as ChatGPT (Achiam et al., 2023) and LLaMA (Touvron et al., 2023), artificial intelligence has advanced significantly, including in robotic task planning. Current language model-based methods can be grounded into two categories (Zhao et al., 2024): LLM-aided planning methods and LLM-native planning methods. LLM-aided methods combine LLMs with classical techniques to improve efficiency and usability. For example, LLMs can translate natural language tasks into formal PDDL problems solved by symbolic planners (Xie et al., 2023; Kambhampati et al., 2024), or construct world models to guide heuristic search (Zhao et al., 2023). LLM-native methods directly generate executable plans without external solvers. Some rely on domain descriptions (e.g., PDDL) as input, while others process natural language or multi-modal signals. Jansen (2020) shows LLMs producing robot instructions from text, and Brohan et al. (2023) integrates images and user prompts for end-to-end planning.

To enhance safety and reliability, Ahn et al. (2022) evaluates and selects feasible LLM-suggested skills at each step, while Lin et al. (2023) extends this to multi-step planning. Other works (Jha et al., 2023; Xu et al., 2024) employ external verifiers to ensure plans satisfy safety constraints expressed in First-Order Logic (FOL) (Barwise, 1977) or Linear Temporal Logic (LTL) (Bauer et al., 2010).

For LRLMs, existing studies (Chen et al., 2025b; Valmeekam et al., 2024; Stechly et al., 2024) primarily focus on replacing the LLMs in prior LLM-based methods with LRLMs, rather than redesigning approaches to leverage the unique reasoning capabilities introduced by LRLMs. While these efforts demonstrate that LRLMs can improve the performance of LLM-based methods, the results remain short of being fully satisfactory, indicating the need for further advancements (Valmeekam et al., 2024).

### 2.2 LRLM EVALUATION

In evaluating LRLMs, several commonly used metrics focus on evaluating both the effectiveness of LRLMs in achieving desired outcomes and their learning efficiency. **Accuracy** or **success rate** measures the proportion of correct outputs. **Pass@k** checks whether at least one correct solution appears within $k$ attempts, and **Cons@k** evaluates the consistency of producing correct or logically coherent outputs across attempts. These metrics primarily focus on evaluating the overall outcomes of the model.

As for benchmarks, in addition to many outcome-oriented benchmarks, such as (Hendrycks et al., 2021; Suzgun et al., 2022; Cobbe et al., 2021), there also exist benchmarks that aim to evaluate the reasoning process itself. (Golovneva et al., 2022; Prasad et al., 2023; Li et al., 2023; Chen et al., 2025a; Bi et al., 2025) assess the model's step-by-step reasoning ability along long reasoning traces. However, these approaches often require training additional evaluators, which introduces additional costs, including manually annotated datasets and substantial computational resources for training. Moreover, the quality of the evaluator directly affects the reliability of the results. In addition, these benchmarks mainly focus on properties such as semantic consistency, logicality, informativeness, and fluency. These criteria fall short in safety-critical contexts that are essential in the motion planning domain. In contrast, our approach overcomes these limitations by eliminating the need for manually annotated datasets or trained evaluators. Instead, it leverages existing formal verifiers for evaluation, with a focus on safety-critical contexts.

Some studies (Lin et al., 2024; Yan et al., 2024; Huang et al., 2025; Chen et al., 2025b; Abacha et al., 2024) evaluate the ability of LRLMs to identify, reflect upon, and correct errors during their reasoning process. For example, FINEREASON (Chen et al., 2025b) is a logic-puzzle benchmark designed to assess whether LRLMs can detect and fix errors in reasoning. It presents models with intermediate states from logic puzzles and asks them to judge whether the reasoning is correct. However, this approach does not directly evaluate the LRLMs' own reasoning process. It mainly measures their ability to act as external discriminators. In contrast, our method directly analyzes how LRLMs handle and correct errors within their own generated reasoning processes.

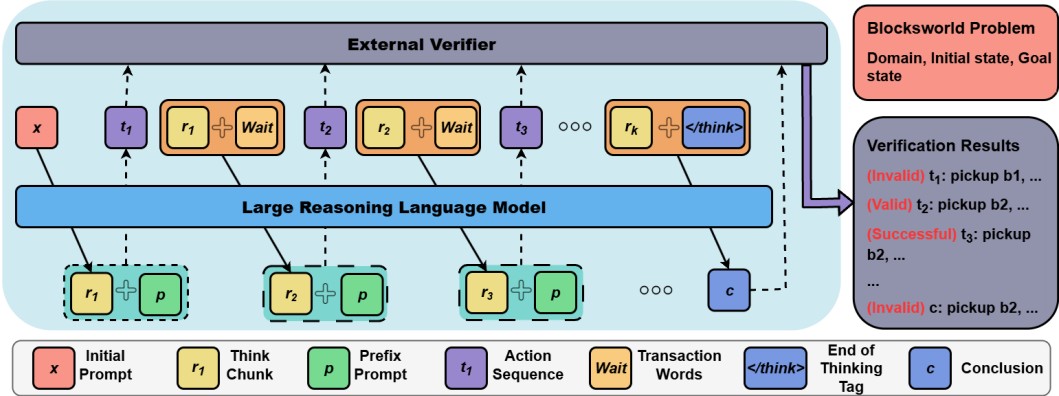

Figure 1: Overall process of our proposed evaluation framework. The process begins with the initial prompt (red cell) being fed into the LRLM. While the LRLM generates tokens sequentially, the framework monitors for the occurrence of transition words, such as 'Wait'. When it is detected, the generation is paused, and the current output is treated as the first reasoning chunk $\mathbf{r}_1$ (yellow cell). This reasoning chunk is then combined with a prefix prompt $\mathbf{p}$ (green cell) that instructs the LRLM to immediately summarize its reasoning into a partial action sequence. The LRLM outputs this sequence $\mathbf{t}_1$ (purple cell), which is sent to an external formal verifier for checking. The verification result is recorded in the verification results section on the right. After that, the LRLM is rolled back to the state at the end of $\mathbf{r}_1$. The transaction word is appended, and the model continues generating to produce the next reasoning chunk $\mathbf{r}_2$. This process is repeated until the final conclusion $c$ (blue cell) is produced, which is also submitted to the external verifier.

## 2.3 LRLM EVALUATION FOR TASK PLANNING

In the planning domain, beyond basic metrics such as success rate, researchers also focus on the efficiency of plan execution and the optimality of the generated plans (Jia et al., 2024; Valmeekam et al., 2023). Other metrics evaluate whether the generated plans are technically sound and practically implementable, meaning whether they satisfy all specified constraints (Xie et al., 2024; Guo et al., 2024). However, existing works, such as (Valmeekam et al., 2024; Stechly et al., 2024), have shown that LRLMs achieve significant improvements over LLMs on classical task planning problems such as PDDL. But the overall performance is still far from satisfactory, and they do not provide a fine-grained analysis of the reasoning processes of LRLMs. The limitation of the LRLM reasoning process in safety-critical problems has not been adequately evaluated. Our work addresses this gap and offers new insights for improving LRLMs.

## 3 METHOD

### 3.1 PRELIMINARIES

**Classic Planning Problems.** We focus on classic planning problems to reveal the ability of LRLMs in safety-critical domains. These problems aim to find a sequence of actions that transforms an initial state into a desired goal state under deterministic and fully observable conditions. Planning Domain Definition Language (PDDL) is a widely used formalism for describing such problems, and our main experiments are conducted on several standard PDDL domains.

A PDDL specification consists of three key components: Domain defines the available action types, their preconditions, and their effects; Initial state specifies the set of facts that are true at the start of the planning problem; Goal describes the conditions that must hold true in the final state. A plan is a sequence of actions that, when applied from the initial state, satisfies all constraints and reaches a state that fulfills the goal conditions. The correctness of a plan can be automatically verified using external validators such as VAL (Howey et al., 2004).

**Large Reasoning Language Models.** Traditional LLMs typically follow a System 1-style generation pattern, which can be represented as:

$$System\ 1 : [Prompt] + [Completion]$$

In contrast, recent LRLMs adapt a System 2-style generation pattern that incorporates an explicit slow-thinking phase, which can be represented as:

$$System\ 2 : [Prompt] + <think> + [Slow-thinking] + </think> + [Conclusion]$$

where $<think>$ and $</think>$ denote the beginning and end of the slow-thinking process, i.e., the reasoning process. During this phase, the LRLM conducts analysis and incrementally reasons through the problem step by step before producing the final answer in the conclusion segment.

We model an LRLM as a next token predictor $LRLM : \mathcal{V}^* \to \mathcal{V}$, where $\mathcal{V}$ is the vocabulary set and $\mathcal{V}^*$ denotes the set of all possible token sequences over $\mathcal{V}$. Given an input sequence, the model predicts the most probable next token. Let $\mathbf{x} = (x_1, ..., x_n) \in \mathcal{V}^*$ represents an input context, with each $x_i \in \mathcal{V}$. We use $[\mathbf{a}, \mathbf{b}]$ to denote the concatenation of two token sequences $\mathbf{a}$ and $\mathbf{b}$.

In the slow-thinking phase, the model first produces a sequence of intermediate reasoning tokens, referred to as a reasoning process, denoted by $\mathbf{r} = (r_1, ..., r_k)$. Each token in this reasoning process is generated auto-regressively based on the input context and all previously generated reasoning tokens: $r_i = LRLM([\mathbf{x}, \mathbf{r}_{<i}])$. After completing the reasoning process, the model enters the conclusion phase, where it generates the final output $\mathbf{y} = (y_1, ..., y_m) \in \mathcal{V}^*$. Each token in the output is produced by conditioning on the input context, the entire reasoning process, and the previously generated output tokens: $y_i = LRLM([\mathbf{x}, \mathbf{r}, \mathbf{y}_{<i}])$.

### 3.2 FINE-GRAINED SAFETY EVALUATION OF REASONING PROCESS

The overall process of our proposed evaluation framework is shown in Fig 1. We use PDDL problems as inputs to the LRLMs and evaluate their ability to generate valid plans that satisfy all given constraints and achieve the goal state. To enable fine-grained evaluation of their reasoning processes, the first challenge is how to segment the reasoning chain into meaningful units that can each be evaluated, since it is infeasible to assess the safety of individual tokens from a planning perspective. Prior studies suggest that the reasoning process can be divided into reasoning chunks based on the occurrence of transition words such as "wait", "alternatively", or "hmm". The appearance of these transition markers typically signals that the LRLM has entered a new stage of its reasoning process. Each reasoning chunk at this stage mainly consists of descriptive natural language rather than well-structured action sequences in a formal format, and thus cannot be directly verified by a formal verifier. However, verification is essential for evaluating task planning tasks, which are inherently safety-critical.

To convert raw reasoning chunks into action sequences that can be formally verified, a straightforward approach is to apply an external natural language processing model to summarize each chunk. However, this will make the verification results dependent on the accuracy of the external summarization model and require additional training efforts. Instead, our method leverages an enforced prefix strategy to prompt the LRLM itself to summarize its own reasoning. Specifically, during the output generation process, when the LRLM produces a reasoning chunk $\mathbf{r} = (r_1, ..., r_k)$ and the next token $r_{k+1}$ is identified as a transition word, we pause the prediction and insert a predefined prefix prompt $\mathbf{p}$, which ask the model to provide a formatted plan that can be in an unfinished state based on the current thinking process. We then ask the model to continue predicting according to: $t_1 = LRLM([\mathbf{x}, \mathbf{r}, \mathbf{p}])$, and get a sequence $\mathbf{t} = (t_1, ..., EOS)$, where $EOS$ is a termination symbol for output and $\mathbf{t}$ is a partial plan in a structured format summarizing the current reasoning chunk and can be directly checked by an external formal verifier. It's important to note that the prefix prompt $\mathbf{p}$ ends with an end of the slow-thinking process tag $</think>$, which prevents the LRLM from continuing its reasoning and instead leads it to directly produce the partial plan corresponding to the current reasoning chunk.

After this summarization step, the model resumes its normal generation from: $r_{k+2} = LRLM([\mathbf{x}, (r_1, ..., r_k, r_{k+1})])$. This approach produces verifiable, structured plans that reflect each reasoning chunk's reasoning while allowing the LRLM to continue its slow-thinking process without disruption. In addition, it does not require the introduction of additional models. For further details of our methods, please refer to the Appendix B.

### 3.3 METRICS

Here we present the results of external verification, which fall into four categories: None, Invalid, Valid, and Successful. None indicates that no verifiable action sequence is produced, either because the reasoning chunk is incomplete or the output format is incorrect. Invalid denotes sequences that violate safety constraints (e.g., attempting to pick up a block while already holding another). Valid refers to sequences that satisfy safety constraints but fail to reach the goal state. Successful means the sequence both satisfies safety constraints and achieves the goal. To ensure statistical reliability, all None cases are excluded from subsequent calculations.

We design four types of metrics to comprehensively evaluate the safety and efficiency of the LRLM reasoning process. The first type includes classical outcome-based metrics commonly used for LLM and LRLM, such as success rate and token usage.

The second type focuses on analyzing inconsistencies between the reasoning process and the final results, measured by two metrics: $R_{suc}F_{nosuc}$ and $R_{nosuc}F_{suc}$. $R_{suc}F_{nosuc}$ represents the proportion of cases where at least one reasoning chunk produces a successful plan during reasoning, but the final result is not successful. $R_{nosuc}F_{suc}$ measures the proportion of cases where the reasoning chunks do not produce a successful plan, yet the final result is successful. These metrics are important because such inconsistencies can reduce the interpretability of LRLMs and undermine users' trust in their outputs. Intuitively, if the final reasoning chunk fails to produce a successful plan, the model's conclusion should not be successful either.

The third type of metrics evaluates the ability of LRLMs to detect and reflect on safety violation errors during their reasoning process. These metrics are crucial for evaluating and improving the capabilities of LRLMs in safety-critical domains. If the model fails to identify safety violation errors in time, it may continue to reason along an incorrect path, which wastes resources such as tokens and computation time. Conversely, triggering reflection on a correct plan can also lead to unnecessary resource consumption and may even reduce the final success rate. To determine whether the LRLM engages in reflection between two consecutive reasoning chunks, we compare the plans generated by these chunks. If the plan in the subsequent reasoning chunk modifies previously generated actions rather than simply appending new ones, we consider this as evidence of safety violation detection and reflection. We formulate this evaluation as a binary classification problem. If the plan generated by the previous reasoning chunk is invalid, we treat it as a true error that requires reflection. If the previous reasoning chunk is valid or successful, we treat it as a case where reflection is not needed. We then use standard binary classification metrics–true positive (TP), false positive (FP), true negative (TN), and false negative (FN)–to measure how well the LRLM can determine whether its own reasoning violates safety during the reasoning process. It's worth noting that in our setting, reasoning chunks labeled as valid are all treated as cases that do not require reflection. In practice, however, LRLM may sometimes revise plans that do not violate safety constraints in order to reach the goal more efficiently. This may slightly affect the experimental results, such as increasing the number of FPs. Nonetheless, such cases are rare because our initial prompt does not explicitly instruct the model to produce the shortest or most efficient action sequence.

The fourth type measures the ability of LRLM to correct its own safety violation errors during the reasoning process. $R_{corr}$ represents the average number of reasoning chunks required to turn the plan from invalid to either valid or successful after correct reflection. Ideally, $R_{corr}$ should be close to 1, meaning that the model can immediately recover from an invalid reasoning chunk in the next reasoning chunk. $\#Tokens$ represents the average number of tokens used to correct each safety violation.

### 3.4 DATASET CONSTRUCTION

As for dataset construction, we use an open-source PDDL problem generator (Seipp et al., 2022) to create planning problems in several classical domains: Blocksworld, Logistics, Depots, Gripper, Ferry, and Miconic. For example, Blocksworld problems involve stacking and unstacking blocks on a table using a robotic arm. The planner must reorder the blocks from an initial state to a target state while respecting constraints such as only moving clear blocks and holding one block at a time. More details about the dataset construction are shown in Appendix A.

|       |     | Blocksworld | Logistics | Depots | Gripper | Ferry | Miconic |
|-------|-----|-------------|-----------|--------|---------|-------|---------|
|       | 7B  | 0.00%       | 0.00%     | 0.00%  | 0.00%   | 5.00% | 20.00%  |
| LLMs  | 14B | 5.00%       | 0.00%     | 0.00%  | 30.00%  | 11.25% | 18.75% |
|       | 32B | 2.50%       | 1.25%     | 1.25%  | 45.00%  | 36.25% | **56.25%** |
|       | 7B  | 18.18%      | 0.00%     | 2.50%  | 45.00%  | 42.50% | 21.25% |
| LRLMs | 14B | 20.91%      | 2.50%     | **7.50%** | **50.00%** | **69.62%** | 41.25% |
|       | 32B | **31.81%**  | **5.00%** | 3.75%  | 26.32%  | 32.50% | 35.00% |

Table 1: Comparison of success rates of LLMs and LRLMs.

For evaluating the generated plans, we use VAL (Howey et al., 2004) to verify their correctness. It is worth noting that both the PDDL problem generator and the VAL verifier we used support the most commonly used PDDL problems, such as Rovers, Satellite, Manufacturing, and so on. This feature allows future researchers to easily expand the dataset using the same tools.

## 4 EVALUATION

### 4.1 BASELINES

To evaluate the impact of model size on the reasoning process, we evaluate three LRLMs with different sizes and their corresponding base LLMs. We intentionally select models with similar architectures and training paradigms to minimize the influence of other factors during evaluation. The LRLMs are **DeepSeek-R1-Distill-Qwen-7B**, **DeepSeek-R1-Distill-Qwen-14B**, and **DeepSeek-R1-Distill-Qwen-32B**. Their corresponding base LLMs are **Qwen2.5-Math-7B**, **Qwen2.5-14B**, and **Qwen2.5-32B**. Future users can easily replace these models with their own to conduct further evaluations using our benchmark framework.

### 4.2 CONFIGURATION

During evaluation, we fix the decoding setting across all models to ensure comparability and reproducibility. The maximum token budget is set to 16384, and the think ratio is set to 0.9. The computations use $bfloat16$, and sampling is disabled ($do\_sample = False$). The initial prompt follows a one-shot template that includes a system instruction, the PDDL domain file, the PDDL problem file, and one worked example. For base LLMs that do not natively produce reasoning traces, we explicitly prompt them to do step-by-step reasoning within $<think>...</think>$ tags first and then give a final plan.

### 4.3 OUTCOME-BASED PERFORMANCE

Tab 1 and Tab 2 present the performance of six models across six PDDL domains. For clarity, the 7B, 14B, and 32B labels in the LLMs columns correspond to the base LLMs with the same parameter sizes introduced in the previous content. The same applies to the LRLMs.

The results show that LRLMs achieve higher success rates than LLMs of the same scale; for instance, the 32B LRLM reaches $31.81\%$ on Blocksworld. However, these rates remain far from satisfactory, and LRLMs consume significantly more tokens, even compared to CoT-prompted LLMs. In some cases, LRLMs exhausted the entire token budget without reaching a conclusion, requiring us to forcibly trigger the conclusion phase. Inspection of their reasoning process suggests they were stuck in confusion, underscoring the need for better inference control methods, especially for real-time domains such as robotics.

### 4.4 INCONSISTENCY BETWEEN REASONING PROCESS AND FINAL RESULTS

Tab 3 presents our evaluation of inconsistencies between the reasoning process and the final results, focusing on LRLMs. The results are measured using two metrics: $R_{suc}F_{nosuc}$ and $R_{nosuc}F_{suc}$. We observe that LRLMs exhibit a considerable degree of inconsistency in task planning. For example, both metrics exceed 10% for the 32B model on the Blocksworld problem. We further find

|  |  | Blocksworld | Logistics | Depots | Gripper | Ferry | Miconic |
|---|---|---|---|---|---|---|---|
| LLMs | 7B | 990 | 212 | 1646 | 828 | 1012 | 462 |
|  | 14B | 962 | 2669 | 1863 | 599 | 444 | 480 |
|  | 32B | 1016 | 842 | 2929 | 589 | 545 | 278 |
| LRLMs | 7B | 7737 | 8405 | 6934 | 3044 | 4441 | 3711 |
|  | 14B | 5801 | 9640 | 7130 | 4159 | 3224 | 3618 |
|  | 32B | 4873 | 9911 | 7771 | 4001 | 2816 | 3830 |

Table 2: Comparison of token usage of LLMs and LRLMs.

|  | $R_{suc}F_{nosuc}$ | | | $R_{nosuc}F_{suc}$ | | |
|---|---|---|---|---|---|---|
|  | 7B | 14B | 32B | 7B | 14B | 32B |
| Blocksworld | 5.74% | 13.75% | 24.32% | 65.00% | 30.43% | 11.42% |
| Logistics | 0.00% | 1.36% | 0.00% | 0.00% | 0.00% | 0.00% |
| Depots | 0.00% | 1.38% | 11.43% | 50.00% | 0.00% | 0.00% |
| Gripper | 0.00% | 70.00% | 92.86% | 44.44% | 20.00% | 0.00% |
| Ferry | 14.29% | 75.00% | 100.00% | 50.00% | 1.82% | 0.00% |
| Miconic | 53.97% | 84.78% | 89.58% | 5.88% | 0.00% | 3.57% |

Table 3: Evaluation of inconsistency between reasoning process and final results

that larger-parameter models, such as the 32B LRLM, exhibit stronger reasoning capabilities but do not necessarily achieve higher consistency between the reasoning process and final results. This is likely due to the lack of explicit emphasis on such consistency during training. This also indicates that larger models do not necessarily yield better performance. For example, the 14B LRLM achieves the highest success rate on the Gripper domain. A possible reason is that the 32B model tends to overthink and deviate from the correct answer, as reflected by its highest $R_{suc}F_{nosuc}$ of 92.86% shown in the table. This finding highlights the need for future research to apply fine-tuning or inference-time techniques to improve consistency, thereby enhancing the transparency and trustworthiness of LRLM-based planning. For some cases, such as the 7B model in Blocksworld, the two metrics differ significantly, which can be attributed to the extremely low success rate according to Tab 1, resulting in too few samples.

### 4.5 ERROR RECOGNITION AND REFLECTION ABILITY

As shown in Tab 4, we formulate the problem of whether LRLMs can recognize and reflect on their own reasoning errors as a binary classification task. This evaluation reveals several noteworthy trends. LRLMs perform well in terms of true possitive rates (TPR), with most models exceeding 90%. However, they still fall short of 100% in most cases, which is essential in safety-critical domains. Further improvement is necessary. In addition, larger models do not necessarily yield higher TPR. For example, in the Gripper domain, the 7B model achieves the highest TPR, outperforming both the 14B and 32B models, which may be due to overthinking in larger models. At the same time, the true negative rates (TNR) remain low across all models, only around 10–20%, indicating that LRLMs often trigger reflection even on correct reasoning processes. One possible cause lies in our evaluation criterion: we determine error recognition and reflection based on whether the LRLM modifies a previously generated action sequence. Although our prompts does not instructed the model to focus on execution time (i.e., the length of the action sequence), we observed that LRLMs frequently revised action sequences even when they were correct, often aiming to shorten the overall plan length. Such behavior increases the number of false positives, which in turn reduces the TNR.

### 4.6 ERROR CORRECTION EFFICIENCY

Tab 5 reports the evaluation results of error correction efficiency, measured by two metrics: $R\_corr$, the proportion of reasoning chunks that successfully corrected prior errors, and #Tokens, the number of tokens consumed during the process. The results show that current LRLMs fall far short of the ideal $R\_corr$ value of 1, indicating limited self-correction capability during reasoning. This limitation leads to substantial token and time consumption. We observe two scenarios in which

|  |  | TP | FP | TN | FN | TPR | TNR |
|---|---|---|---|---|---|---|---|
| Blocksworld | 7B | 511 | 45 | 222 | 27 | 91.90% | 10.84% |
| | 14B | 543 | 42 | 400 | 45 | 92.82% | 10.11% |
| | 32B | 492 | 32 | 411 | 52 | **93.89%** | **11.23%** |
| Logistics | 7B | 71 | 1 | 89 | 1 | **98.61%** | 1.11% |
| | 14B | 427 | 29 | 132 | 17 | 93.64% | **11.41%** |
| | 32B | 209 | 17 | 73 | 7 | 92.47% | 8.75% |
| Depots | 7B | 107 | 3 | 65 | 0 | **97.20%** | 0.00% |
| | 14B | 408 | 33 | 125 | 12 | 92.51% | 8.76% |
| | 32B | 387 | 28 | 100 | 14 | 93.25% | **12.28%** |
| Gripper | 7B | 7 | 0 | 14 | 5 | **100.00%** | **26.31%** |
| | 14B | 33 | 5 | 106 | 24 | 86.84% | 18.46% |
| | 32B | 6 | 1 | 52 | 8 | 85.71% | 13.13% |
| Ferry | 7B | 391 | 24 | 113 | 15 | 94.21% | **11.71%** |
| | 14B | 133 | 6 | 399 | 31 | 95.68% | 7.21% |
| | 32B | 121 | 0 | 322 | 14 | **100.00%** | 4.16% |
| Miconic | 7B | 44 | 0 | 432 | 29 | **100.00%** | 6.29% |
| | 14B | 22 | 1 | 607 | 57 | 95.65% | **8.58%** |
| | 32B | 15 | 0 | 678 | 22 | **100.00%** | 3.14% |

Table 4: Error recognition and reflection ability evaluation results

|  | $R_{corr}$ | | | $\#Tokens$ | | |
|---|---|---|---|---|---|---|
| | 7B | 14B | 32B | 7B | 14B | 32B |
| Blocksworld | 2.88 | 3.59 | 3.72 | 520 | 572 | 641 |
| Logistics | 1.52 | 2.54 | 2.35 | 513 | 616 | 753 |
| Depots | 7.40 | 2.74 | 3.38 | 1455 | 438 | 673 |
| Gripper | 1.57 | 1.65 | 2.70 | 166 | 240 | 402 |
| Ferry | 2.07 | 2.32 | 2.49 | 329 | 413 | 344 |
| Miconic | 1.64 | 2.00 | 1.36 | 267 | 397 | 178 |

Table 5: Evaluation of error correction efficiency

$R_{corr}$ approaches 1. The first occurs when the LRLM's success rate is extremely low, leading to many unverifiable outputs, as in the case of the 7B model on the Logistics domain (1.52%). The second occurs when the success rate is very high, where the LRLM demonstrates strong problem understanding and can quickly repair safety violations, as seen with the 7B and 14B models on the Gripper domain (1.57% and 1.65%). Overall, these findings highlight the urgent need to improve LRLMs' self-correction mechanisms, which would not only enhance reasoning safety but also reduce unnecessary token consumption.

## 5 CONCLUSION

This paper studies the fine-grained safety evaluation of the LRLMs for safety-critical domains. We propose a framework that segments reasoning into chunks, summarizes them into explicit plans, and verifies them against safety constraints. We also design a set of metrics and construct a dataset to systematically evaluate the reasoning behavior of LRLMs. Experiments on PDDL-based tasks confirm both the advantages and the limitations of LRLMs. While LRLMs outperform standard LLMs, they often generate unsafe or inconsistent reasoning steps that reduce reliability in safety-critical domains. Moreover, their ability to detect and correct safety violation errors during the reasoning process is limited. These findings highlight the need for new fine-tuning strategies or inference-time computation methods to enhance LRLMs and support their deployment in real-world safety-critical applications.

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

## A  DATASET CONSTRUCTION

As for dataset construction, we use an open-source PDDL problem generator Seipp et al. (2022) to create planning problems from three classical domains: Blocksworld, Logistics, Depots, Gripper, Ferry, and Miconic.

**Blocksworld** involves stacking and unstacking blocks on a table using a robotic arm. The planner must reorder the blocks from an initial configuration to a target configuration while respecting constraints such as only moving clear blocks and holding one block at a time. Specifically, for each problem instance, the number of actions is set to 4, and the number of blocks varies between 2 and 10. In total, 110 problem instances are generated randomly.

**Logistics** focuses on transporting packages between locations using trucks and airplanes. The planner must coordinate load, unload, drive, and fly actions while optimizing the delivery of multiple packages across cities and airports. Specifically, for each problem instance, the number of cities ranges from 2 to 5, with each city containing 3 to 8 locations. The number of packages varies between 4 and 15, the number of airplanes between 1 and 6, and the number of trucks is set equal to the number of cities. In total, 80 problem instances are generated randomly.

**Depots** combines elements of both Blocksworld and Logistics. It requires managing cranes to load and unload packages from trucks while organizing the storage of packages in stacks at depots, making it more complex and resource-constrained than the other two domains. Specifically, for each problem instance, the number of depots is set between 1 and 6, distributors between 2 and 6, trucks between 2 and 6, pallets between 3 and 20, crates between 2 and 20, and hoists between 3 and 15. In total, 80 problem instances are generated randomly.

**Gripper** involves moving balls between rooms using a robot equipped with two grippers. The planner must transfer all balls from the initial room to the target room, while respecting constraints such as the robot's location, the availability of grippers, and the carrying capacity. Specifically, for each problem instance, the number of balls varies between 1 and 20. In total, 20 problem instances are generated randomly.

**Ferry** involves transporting cars between multiple locations using a ferry. The planner must load and unload cars onto the ferry and move the ferry between locations to reach the target goal, subject to constraints on ferry capacity and location. For each problem instance, the number of locations is set between 2 and 5, and the number of cars between 1 and 4. In total, 80 problem instances are generated randomly.

**Miconic** models an elevator control problem where passengers must be transported between floors. The planner must move the lift and stop at appropriate floors so that each passenger boards at their origin floor and debarks at their destination floor, ensuring that all passengers are eventually served. For each problem instance, the number of floors is set between 2 and 4, and the number of passengers between 1 and 4. In total, 80 problem instances are generated randomly.

Among all these tasks, Logistics and Depots are the most challenging, as they involve a larger number of actions and objects.

## B  FINE-GRAINED SAFETY EVALUATION OF REASONING PROCESS

During the fine-grained safety evaluation of the reasoning process, we leverage several important prompts to lead the LRLMs, including the initial prompt and the prefix prompt.

As for the initial prompt, we employ a one-shot strategy that supplies the LRLM with essential task information (such as the domain.pddl file) together with an illustrative problem and its corresponding correct solution. The format of initial prompt is shown as bellow:

```
You are a motion planner for PDDL planning problems.
Now we consider the {{domain_name}} planning problem.

Here is the domian.pddl file of this blocksworld planning problem:

{{domain.pddl}}
```

```
Here is a simple problem and it's answer:

{{example_problem.pddl}}

{{answer}}

Now please give me the result of the new {{domain_name}} problem below.
First, provide detailed reasoning inside the <think></think> tags, and
    then give the final answer outside the tags.
The final answer's format should be the same as the example solution.

Here is the new {{domain_name}} problem:

{{problem.pddl}}
```

Listing 1: Initial prompt format

where $\{\{\cdot\}\}$ denotes the part to be replaced according to the specific problem. For example, $\{\{domain.pddl\}\}$ indicates that it should be replaced with the content of the corresponding domain.pddl file for the current problem.

The following provides an example of the initial prompt for a Blocksworld problem.

```
You are a motion planner for PDDL planning problems.
Now we consider the blocksworld planning problem.

Here is the domian.pddl file of this blocksworld planning problem:

(define (domain blocksworld)
  (:requirements :strips)
(:predicates (clear ?x)
             (on-table ?x)
             (arm-empty)
             (holding ?x)
             (on ?x ?y))

(:action pickup
  :parameters (?ob)
  :precondition (and (clear ?ob) (on-table ?ob) (arm-empty))
  :effect (and (holding ?ob) (not (clear ?ob)) (not (on-table ?ob))
               (not (arm-empty))))

(:action putdown
  :parameters  (?ob)
  :precondition (holding ?ob)
  :effect (and (clear ?ob) (arm-empty) (on-table ?ob)
               (not (holding ?ob))))

(:action stack
  :parameters  (?ob ?underob)
  :precondition (and (clear ?underob) (holding ?ob))
  :effect (and (arm-empty) (clear ?ob) (on ?ob ?underob)
               (not (clear ?underob)) (not (holding ?ob))))

(:action unstack
  :parameters  (?ob ?underob)
  :precondition (and (on ?ob ?underob) (clear ?ob) (arm-empty))
  :effect (and (holding ?ob) (clear ?underob)
               (not (on ?ob ?underob)) (not (clear ?ob)) (not (arm-empty)
    ))))

Here is a simple problem and it's answer:

Here is a simple example:
```

```
(define (problem BW-sample-0)
(:domain blocksworld-4ops)
(:objects b1 b2 b3)
(:init
(arm-empty)
(on b1 b2)
(on b2 b3)
(on-table b3)
(clear b1)
)
(:goal
(and
(on b3 b2)
(on b2 b1)
(on-table b1)
)))

Here is a answer of this example problem:
START-PLAN
1. unstack b1 b2
2. put-down b1
3. unstack b2 b3
4. put-down b2
5. pick-up b2
6. stack b2 b1
7. pick-up b3
8. stack b3 b2
END-PLAN

Now please give me the result of the new blocksworld problem below.
First, provide detailed reasoning inside the <think></think> tags, and
    then give the final answer outside the tags.
The final answer's format should be the same as the example solution.

Here is the new blocksworld problem:

(define (problem BW-rand-6)
(:domain blocksworld-4ops)
(:objects b1 b2 b3 b4 b5 b6 )
(:init
(arm-empty)
(on-table b1)
(on b2 b3)
(on-table b3)
(on b4 b2)
(on b5 b6)
(on-table b6)
(clear b1)
(clear b4)
(clear b5)
)
(:goal
(and
(on b1 b5)
(on b2 b6)
(on b4 b2)
(on b5 b4))
)
)
```

Listing 2: An example of the initial prompt

Another important prompt is the prefix prompt, which guides the LRLM to generate an incomplete plan based on the current reasoning process. The detailed content is shown below.

```
</think>. So far, I get part of the final plan.
I will write it in this format:
START-PLAN
1. ...
2. ...
3. ...
END-PLAN
Here is part of my current plan:
```

Listing 3: Prefix prompt

The leading $</think>$ forces the termination of the current reasoning process and the transition to the conclusion stage, requiring the LRLM to produce a partial plan based on the reasoning generated so far.

## C   THE USE OF LARGE LANGUAGE MODELS (LLMS)

We use LLMs to assist with writing tasks, including grammar checking and improving readability.

