# OpenReview forum: "Towards Fine-grained Evaluation of Large Reasoning Language Models in Task Planning"
_ICLR.cc/2026/Conference — ICLR 2026 Conference Withdrawn Submission_

### Official Review · Reviewer_m79v · 2025-10-15

**Soundness:** 1
**Presentation:** 2
**Contribution:** 1
**Rating:** 0
**Confidence:** 4

**Summary:**

# Summary
## What is the problem solved? Is it a known problem?
The paper deals with evaluating the output of reasoning models when they are tasked with plan generation in classical planning. I addition to the well-established metrics of evaluating planners outcomes (e.g., measuring success rate, known in the planning community as coverage), the authors propose an evaluation of the observation of the planning process of reasoning models, i.e., the reasoning output together with the final output.

## How is it solved in the literature and how is the current approach different?
This is not a common task in the literature, as it makes little sense. Language models are notoriously bad at plan generation in the domain-independent setting, not just because the language models are not yet ready or need to be more fine-tuned, but because they are just not the right tool for the task (see, e.g., [1, 2, 3]).

While they can show a non-zero performance on easiest planning tasks they were trained on, and generalize to repeating the same pattern (like in Gripper, where the plans all look like pick, pick, move, drop, drop, move, repeated as many times as needed), or replacing objects with similar ones (like in BlocksWorld, where there are only a few plans patterns), moving to recent (e.g., from 2023 planning competition) or better yet unseen domains of higher than P complexity class makes the performance go down to 0. This is due to a simple fact: even if you restrict yourself to applicable sequences of actions (and language models cannot), their number grows astronomically, while the portion of these that actually achieve the goal is microscopic. In order to find solutions for these tasks, modern automated planners perform a systematic (heuristic) search, exploring millions and millions of states on their way to the goal. They know how to avoid exploring the entire state space, focusing on portions of the space that are deemed more promising by the heuristic functions, and exploring only the "valid" options -- successors achieved via applicable actions.  Language models don't work that way (they cannot guarantee that only sequentially applicable actions are produced at each step) and if they did, it would be sufficient to just measure their coverage (aka success rate).

It is not clear what is a purpose of evaluating the intermediate output we obtain from the reasoning model during the process of plan generation, since the models themselves are treated as black boxes anyway, so the research direction is not well-motivated. If the motivation is to improve the reasoning models in the future, the authors should  at least mention how they propose improving them and how these metrics help with that.
The authors mention finetuning, but finetuning can only help in a domain-dependent planning setting, by training the model to observe more plans that are similar to the plans you are going to ask it to produce. It does not work in the domain-independent setting, where you have no idea what will be the next planning problem you are asked to solve.

Most importantly, there seems to be a hidden assumption that the reasoning of the language models is forward chained from the initial state, which roughly corresponds to a plan. There is no justification provided for such an assumption. What if the reasoning is backward chained from the goal (e.g., action a achieves this atom in the goal, action b achieves some precondition of a, action c achieves another atom in the goal and precondition of b, not ruining any preconditions of a)


# Significance

There is little surprise in the results of the paper, given that the literature already holds the same opinion about the plan generation capabilities of language models. The dataset constructed for the empirical evaluation is on domains that the instances and plans for are frequent in the training data of language models and therefore of less practical use.

# Soundness

There is no theory, but some of the choices made raise concerns. For instance, in the metrics section, third type of metrics makes unsupported assumptions that just don't hold in planning in general, and violated even in the simplest domains the authors experiment with, namely Gripper. The authors say "If the plan in the subsequent reasoning chunk modifies previously generated actions rather than simply appending new ones, we consider this as evidence of safety violation detection and reflection. We formulate this evaluation as a binary classification problem." There is no way of detecting whether the subsequent reasoning step comes to replace the previous one or append to it. In fact, it can even be identical to the previous one - e.g., abd comes after abc or even ab comes after ab can actually happen in planning.

# Novelty

As the aim is to evaluate more nuanced abilities of language models, such as applicability and goal achievement, it is worth looking at a very related and completely ignored literature on evaluating exactly these abilities instead of plan generation [4,5,6,7]. These papers evaluate action applicability, how a state changes as a result of applying an action, goal achievement, etc.


# Scholarship

[1] Abbe et al., NeurIPS 2024, How Far Can Transformers Reason? The Globality Barrier and Inductive Scratchpad
[2] Shojaee et al., Arxiv 2025, The Illusion of Thinking: Understanding the Strengths and Limitations of Reasoning Models via the Lens of Problem Complexity
[3] Valmeekam et al., NeurIPS 2023, On the Planning Abilities of Large Language Models - A Critical Investigation
[4] Handa, D.; Dolin, P.; Kumbhar, S.; Baral, C.; and Son, T. C. ICLR 2025. ActionReasoningBench: Reasoning about Actions with and without Ramification Constraints
[5] He et al., ACL 2023, Exploring the Capacity of Pretrained Language Models for Reasoning about Actions and Change
[6] Kokel et al., AAAI 2025, ACPBench: Reasoning about Action, Change, and Planning
[7] Kokel et al., LM4Plan@AAAI 2025, ACPBench Hard: Unrestrained Reasoning about Action, Change, and Planning


# Clarity

The setting that the paper deals with is not clear. The authors say that they deal with the classical setting and clarify that they mean by that deterministic and fully observable. And then they mention PDDL, but PDDL covers quite a large fragment of planning formalisms. Even if we restrict ourselves to deterministic (action effects) and full observability (and you still should mention closed world assumption), PDDL supports both propositional and numeric planning. In planning literature, classical usually refers to the propositional setting.
It is not clear what sub-fragment is supported in this work. For instance, even the propositional (classical) setting includes ADL features, axioms and derived predicates, etc. The authors should be clear about the setting they investigate in this work.

The paper uses ill-defined concepts such as "safety constraints", "reliable", "interpretability", "transparency" and more. It also often uses the term "plan" in a loose sense.
The field of planning has well-established and precise jargon. An action is applicable in a state if its preconditions hold in a state, a sequence of actions a_1, ..., a_n is applicable in a state s_0 if a_i is applicable in s_{i-1}, s_i is the result of applying a_i in s_{i-1}, and it is a plan if the end state s_n is consistent with the goal. So, a plan is an applicable action sequnece that achieves a goal state. If an action sequence is not applicable or does not achieve a goal state, it is not a plan and should not be called that, it is confusing.
So, I would suggest to use the terms "applicable" or "sequentially applicable" and "goal reaching" or "goal achieving" as well as "action sequnence" and "plan" where appropriate.


# Evaluation and Reproducibility

The authors evaluate on 6 classical planning domains: BlockWorld, Logistics, Depots, Gripper, Ferry, and Miconic. The choice of domains is not well motivated. These domains are among the most well-known, as they are very simple. In fact, they are all polynomial for plan generation, and there exist simple policies that solve any task in these domains (see the vast literature on generalized planning). Also, these are the domains that I would have most concerns about language models memorization.

The planning community maintains a large variety of PDDL domains with their generators (see https://github.com/AI-Planning/pddl-generators ), the newer domains can be found via the respective International Planning Competition (IPC) websites. For instance, IPC 2023 classical track https://ipc2023-classical.github.io/
featured the following domains:
 * Folding
 * Labyrinth
 * Quantum Circuit Layout Synthesis
 * Recharging Robots
 * Ricochet Robots
 * Rubik's Cube
 * Slitherlink

If you conducted your experiments on these domains, I bet you would see a very different picture. Some of these still have generalized policies (e.g., Rubik's cube), and it mihgt be worth also to obfuscate the evaluated domains.

# Comments on Technical Details

1. Intro, line 43: "ensure logical consistency" -- these models cannot *ensure* logical consistency, you might want to use a different verb.
2. Related work, line 119: the citation Kambhampati et al., 2024 is wrong here, possibly you meant Guan et al., NeurIPS 2023 Leveraging Pre-trained Large Language Models to Construct and Utilize World Models for Model-based Task Planning  by the same authors?
3. Section 2.2, first paragraph, lines 138 - 143, citation(s) for these metrics are missing.
4. Section 2.3 line 191: "optimality of the generated plans" optimality is not the right word here, but possibly you meant quality? Optimal planning deals with finding a provably optimal plan, if a plan that was found is not optimal, it is not a solution. This is due to the fact that checking whether a sequence of actions is a plan is poly-time, while checking whether it is optimal is PSPACE-hard.
5. Section 2.3 lines 195-196: "classical task planning problems such as PDDL" - PDDL is a language for capturing planning tasks (classical, but also way beyond classical), not a problem.
6. Line 206: The correct term is "classical", not "classic". I would also strongly encourage you to be more formal in this paragraph.
7. Line 242: "the goal state" -> "a goal state", there can be more than one (and often is).
8. Line 262: It is not clear how the "partial plans" can be validated with VAL. While I can imagine that the first part can be validated by just providing it to the validator together with the original domain and problem files, but what do you do with the next one, especially if the first one is not sequentially applicable?
9. Lines 335 - 337: "It is worth noting that both the PDDL problem generator and the VAL verifier we used support the most commonly used PDDL problems, such as Rovers, Satellite, Manufacturing, and so on." This sentence does not make sense to me. How do the PDDL problem generators support most commonly used PDDL problems? The generators are domain-specific and were created specifically for each domain to generate probems in that domain in PDDL format. VAL is a validator for PDDL, that supports PDDL 2.2 (Edelkamp and Hoffmann, Technical report 2004, PDDL2.2: The Language for the Classical Part of the 4th International planning Competition).

**Strengths:**

1. The paper suggests novel metrics for evaluating the reasoning output of reasoning models in addition to the final output.
2. The paper performs experimental analysis using a language model of three sizes, based vs their DeeSeek-R1 distilled versions as a reasoning model
3. The paper uses 6 PDDL domains.

**Weaknesses:**

1. There seems to be a hidden assumption that the reasoning of the language models is forward chained from the initial state, which roughly corresponds to a plan. There is no justification provided for such an assumption.
2. There is no justification provided for evaluating the reasoning traces of reasoning models. This is an intermediate output, not the one that is used to come up with a final response plan.
3. There is no explanation how these particular metrics can help improve the reasoning models capabilities in plan generation.
4. The experimental evaluation is using very simple domains that are also very popular and are out there for many years, together with many instances and their plans, raising memorization concerns. No ablation studies are performed to obfuscate the domains.
5. The paper is not easy to read, it uses uncommon in planning field terminology and ill-defined terms. It does not provide a formal definition for the problem solved or for the planning formalism it focuses on, and only describes it in vague terms.

**Questions:**

1. What evidence can you suggest that reasoning models can significantly improve their plan generation abilities?
2. How do you envision the proposed metrics contributing to improving the plan generation abilities?
3. What if the reasoning does not correspond to a forward chaining plan? What if it is backward chained from the goal?

---

### Official Review · Reviewer_vFxG · 2025-10-28

**Soundness:** 3
**Presentation:** 3
**Contribution:** 2
**Rating:** 4
**Confidence:** 4

**Summary:**

Paper proposes a fine-grained evaluation framework for assessing the safety of Large Reasoning Language Models (LRLMs) in task planning domains. The authors segment LRLM reasoning processes into chunks using transition word detection, then use enforced prefixes to prompt the model to summarize each chunk into verifiable action sequences. These summaries are checked against safety constraints using external PDDL verifiers. The framework is evaluated on six PDDL domains (Blocksworld, Logistics, Depots, Gripper, Ferry, Miconic) using DeepSeek-R1-Distill models of varying sizes. Key findings include: (1) significant inconsistencies between reasoning processes and final outputs (up to 92.86%), (2) high true positive rates (90%+) but low true negative rates (10-20%) for error detection, and (3) poor self-correction efficiency requiring 2-7 reasoning chunks to recover from errors. Evaluation metrices are nicely explained but most of the insights are very much known in the LLM (upto certain extent in planing community) community and this paper confirms that in the planning domain.

**Strengths:**

1. Timely and Important Problem: Addresses a critical gap in evaluating LRLM safety and reliability in safety-critical domains, which is increasingly relevant as reasoning models are deployed in real-world applications.

2. Novel Process-Level Metrics: Introduces meaningful chunk-level evaluation metrics (R_suc_F_nosuc, R_nosuc_F_suc, R_corr) that go beyond traditional outcome-based measures and capture reasoning-conclusion inconsistencies.

3. Practical Evaluation Framework: The chunk-based verification approach using enforced prefixes and rollback techniques is creative and could be adapted to other domains with formal verifiers.

4. Comprehensive Experimental Design: Systematic evaluation across multiple model sizes (7B, 14B, 32B but of same model kind), six PDDL domains, and comparison between base LLMs and LRLMs provides useful empirical data and reaffirm already known issues of the reasoning model.

5.Quantification of Known Issues: Provides concrete numbers for previously suspected problems (e.g., 5-10x token overhead, low success rates, self-correction inefficiency).

6. Clear Presentation of Results: Tables and metrics are well-organized and findings are generally communicated clearly.

**Weaknesses:**

1. The chunk segmentation method is heuristic and under-validated; results may depend heavily on chosen transition words.

2. Heavy dependence on the model’s self-summarization introduces a confound—failures might reflect poor summarization, not unsafe reasoning.
3. Limited model diversity (mostly DeepSeek-R1 and Qwen-based) restricts the generality of conclusions.For example, thnking process of different model varies and it might be possible that Llama will be better in this direction. Since this paper care more for the reasoning trace, so it doesnt matter the underlying architecture of language model provided they perform consistent CoT.

4. Missing ablation studies (e.g., segmentation variations, external summarizer vs. self-summarizer, prompt sensitivity).

5. Statistical analysis and variance reporting are minimal; unclear if results are robust.

6. Appendix B only provides detail about prompt rather than methodology. But main text refer to this section for more details about method.

7. Claims such as “overthinking leads to more inconsistency” are not causally demonstrated, just observed.

8. Incomplete Insights:
 - Also, no human evaluation or inter-annotator agreement studies (No comparison with human expert reasoning to validate whether these failure rates are actually problematic)
 - Most findings confirm prior work (Valmeekam et al., 2024 already showed LRLMs perform poorly on planning)

**Minor points**

1. Presentation Issues:
 - Formal LRLM notation in Section 3.1 is introduced but underutilized
- Missing implementation details (how is KV cache rollback performed?)
- Figure 1 could be more concise

**Questions:**

1. Are inconsistencies due to training, architecture, or inference dynamics of underlying reasoning model?

2. How robust is your segmentation to different “transition word” lists? Can you show results using an alternative or automatic segmentation strategy? These transition words varies by the underlying model.

3. Have you compared model-generated summaries with human-written or external summaries to see how much summarization quality affects verification accuracy?

4. Did authors test other LRLMs (beyond DeepSeek/Qwen) or baselines like standard CoT prompting or Veriplan-based checks?

5. Could you quantify per-instance variance or confidence intervals for your metrics to support the observed trends?

6. The paper claims that large models “overthink”: can you provide controlled evidence (e.g., limiting think tokens) to support this interpretation?

7. Regarding rollback implementation: Exactly how is KV cache rollback implemented? At what granularity (token, layer)? What are the computational costs?

---

### Official Review · Reviewer_rJ9u · 2025-10-29

**Soundness:** 2
**Presentation:** 2
**Contribution:** 2
**Rating:** 4
**Confidence:** 3

**Summary:**

This paper identifies the lack of comprehensive evaluation methods for Large Reasoning Language Models (LRLMs) in safety-critical task planning. To address this gap, the authors propose a fine-grained and systematic evaluation framework that segments the reasoning process into multiple chunks, summarizes each chunk, and verifies their compliance with safety constraints using an external verifier. The framework enables in-depth analysis of whether LRLMs violate safety constraints during reasoning and evaluates their ability to detect and correct such errors. Experiments on various PDDL-based planning domains reveal that LRLMs often show inconsistencies between their reasoning process and final outputs, and still have limited self-correction capabilities. These findings highlight important directions for improving the reliability and safety of LRLMs in real-world applications.

**Strengths:**

1.Sound methodology: The paper introduces a novel framework that segments the reasoning process of LRLMs into multiple chunks and evaluates each chunk for safety. This approach allows for a more detailed and systematic analysis of reasoning errors and safety violations compared to traditional outcome-based metrics.
2.Comprehensive Evaluation Metrics: In addition to classic success rate and token usage, the framework proposes new metrics to assess the consistency between reasoning steps and final results, as well as the model’s ability to detect and correct its own errors. This provides a more holistic and informative evaluation of LRLM reasoning behavior.

**Weaknesses:**

1.Experimental Scope and Diversity: While the experiments cover several classical PDDL domains, the evaluation is limited to these specific structured planning tasks. The generalizability of the proposed framework to more open-ended or less formally defined domains has not been explored.
2.Clarity: While the step-by-step introduction in the methods section is clear, the main points of each component are not sufficiently highlighted, making it harder for readers to grasp the central ideas. To improve organizational effectiveness and readability, small paragraph headings or subheadings (e.g., “Chunk Segmentation”, “Chunk Summarization”, “External Verification”) could be added to clearly separate each step.
3.Limited baselines: The experimental evaluation includes a limited number of models, which may affect the generalizability of the findings. With only a small number of LRLMs and LLMs analyzed, it is challenging to identify robust patterns or draw broadly applicable conclusions that could inform future improvements.

**Questions:**

1. Some notations and symbols (e.g., R_{corr} in Section 4.6) could be better organized and consistently presented.

---

### Official Review · Reviewer_mAGm · 2025-10-30

**Soundness:** 3
**Presentation:** 2
**Contribution:** 2
**Rating:** 4
**Confidence:** 3

**Summary:**

This paper presents a fine-grained safety evaluation framework for LRLMs in robotic task planning. Unlike prior outcome-based evaluations, the framework analyzes reasoning processes by segmenting them into interpretable chunks, summarizing each into verifiable plans, and using an external verifier to check safety compliance. Experiments show that LRLMs frequently generate unsafe or inconsistent reasoning and show limited self-correction. The main contribution of the work is a systematic framework combining reasoning segmentation, summarization, and verification, along with a benchmark and safety metrics for assessing LRLM limitations.

**Strengths:**

1. The paper addresses a timely and important problem, including the safety of reasoning processes in LRLM.
2. The study provides a necessary research direction for deploying LRLMs safely in safety-critical robotic planning tasks.
3. It makes a meaningful contribution by shifting the paradigm of LRLM evaluation from outcome-based assessment to process-oriented evaluation, as it focuses on reasoning-result inconsistency, error detection, and correction capabilities.
4. The paper provides detailed descriptions of dataset construction and example prompts, which will help future researchers reproduce and extend the work.

**Weaknesses:**

1. The proposed framework mainly focuses on evaluating the reasoning safety of LRLMs, but it does not yet incorporate mechanisms for model improvement or control. In its current form, it is more like a valuable diagnostic tool rather than a method that actively enhances model safety. Therefore, this reviewer feels that the overall study maturity remains at an early stage. Extending the framework with feedback or intervention capabilities could strengthen its practical impact and contribution to advancing LRLM robustness.
2. Although the paper mentions the use of VAL (Howey et al., 2004), the integration details are somewhat unclear. Clarifying how the verifier is incorporated into the framework, how verification results interact with LRLM outputs, and how success or failure is determined would be important. It would also be helpful to explain whether VAL can process partial or incomplete plans generated by the model.
3.This paper could have been better if it had made more use of visualizations such as graphs or charts. Figure 1 provides an overview of the proposed pipeline, but it is the only visualization; the rest of the paper relies mainly on text and tables, which reduces readability. This reviewer believes that incorporating additional visualizations (such as graphs or charts illustrating per-chunk verification outcomes, inconsistency trends, or reflection frequencies) would make the analysis more comprehensible and effectively convey the "fine-grained" nature of the proposed evaluation.
4. The notion of safety in this work is defined mainly in terms of satisfying preconditions in PDDL, which does not account for richer physical or spatial constraints encountered in real-world robotic settings (e.g., collision avoidance, restricted areas, resource conflicts). Broadening this definition or discussing how the framework could be extended to handle such cases would enhance its generalizability and practical relevance.
5. While the experimental results are interesting, the paper provides limited discussion of their underlying causes. In particular, the finding that larger models sometimes perform worse or show greater reasoning result is intriguing but not sufficiently analyzed. It would strengthen the work to explore potential contributing factors such as reasoning-chain length, attention drift, or hallucination, and to discuss possible mitigation strategies.
6. Overall, the paper raises an important problem and introduces a promising direction, but it currently lacks analytical depth and practical impact to justify acceptance.

**Questions:**

1. Beyond using transition words, were other segmentation criteria explored or considered? If not, how sensitive do the results appear to be to the current segmentation strategy?
2. What is the computational overhead of the proposed framework, and how does this cost scale with the number of reasoning chunks?
3. What is the total end-to-end runtime of the evaluation framework for a typical planning problem, including all iterative generation and verification steps?
4. Why did the Depots task require significantly more tokens with the 7B model than with the 14B and 32B models, and compared to other tasks?
5. What are the key criteria for selecting an external verifier, and how would the evaluation results be impacted by using an alternative to VAL, especially one that might better handle partial plans or richer PDDL constraints?
6. What was the rationale behind selecting the 6 PDDL problems? Do the authors believe that these problems are sufficient to validate the proposed method?
7. The segmentation approach relies on linguistic cues to identify reasoning boundaries. How robust is this method across different models that may exhibit varying reasoning styles or patterns?
8. The paper mentions that the code and dataset will be released, but details are limited. Could the authors provide more information on the planned release format and timeline, along with the dataset’s total size and number of tasks?

---

### Note · Authors · 2025-11-12

I have read and agree with the venue's withdrawal policy on behalf of myself and my co-authors.